# Incretins-Based Therapies and Their Cardiovascular Effects: New Game-Changers for the Management of Patients with Diabetes and Cardiovascular Disease

**DOI:** 10.3390/pharmaceutics15071858

**Published:** 2023-07-01

**Authors:** Federico Bernardini, Annunziata Nusca, Federica Coletti, Ylenia La Porta, Mariagrazia Piscione, Francesca Vespasiano, Fabio Mangiacapra, Elisabetta Ricottini, Rosetta Melfi, Ilaria Cavallari, Gian Paolo Ussia, Francesco Grigioni

**Affiliations:** Unit of Cardiac Sciences, Department of Medicine, Campus Bio-Medico University of Rome, 00128 Rome, Italy; f.bernardini@unicampus.it (F.B.); federica.coletti@unicampus.it (F.C.); ylenia.laporta@unicampus.it (Y.L.P.); mariagrazia.piscione@unicampus.it (M.P.); francesca.vespasiano@unicampus.it (F.V.); f.mangiacapra@policlinicocampus.it (F.M.); e.ricottini@policlinicocampus.it (E.R.); r.melfi@policlinicocampus.it (R.M.); i.cavallari@policlinicocampus.it (I.C.); g.ussia@policlinicocampus.it (G.P.U.); f.grigioni@policlinicocampus.it (F.G.)

**Keywords:** atherosclerosis, type 2 diabetes mellitus, GLP-1 receptor agonists, DPP-4 inhibitors, incretins

## Abstract

Atherosclerosis is the leading cause of death worldwide, especially in patients with type 2 diabetes mellitus (T2D). GLP-1 receptor agonists and DPP-4 inhibitors were demonstrated to play a markedly protective role for the cardiovascular system beyond their glycemic control. Several cardiovascular outcome trials (CVOT) reported the association between using these agents and a significant reduction in cardiovascular events in patients with T2D and a high cardiovascular risk profile. Moreover, recent evidence highlights a favorable benefit/risk profile in myocardial infarction and percutaneous coronary revascularization settings. These clinical effects result from their actions on multiple molecular mechanisms involving the immune system, platelets, and endothelial and vascular smooth muscle cells. This comprehensive review specifically concentrates on these cellular and molecular processes mediating the cardiovascular effects of incretins-like molecules, aiming to improve clinicians’ knowledge and stimulate a more extensive use of these drugs in clinical practice as helpful cardiovascular preventive strategies.

## 1. Introduction

Glucagon-like peptide-1 receptor agonists (GLP1-RA) and dipeptidyl peptidase-4 inhibitors (DPP4i) were recently introduced as novel classes of drugs for treating patients with type 2 diabetes (T2D). However, several cardiovascular preclinical and clinical studies, originally designed to investigate their safety in clinical practice, early reported an impressive and effective performance of these agents in reducing cardiovascular (CV) risk, and thus attenuated incidence of acute myocardial infarction or stroke in patients with T2D receiving these drugs associated with lower cardiovascular mortality [1,2,3,4,5,6,7,8,9,10,11,12,13]. Following this evidence, both cardiological and endocrinological guidelines [14,15] now recommend GLP-1 RA for managing patients with both T2D and high atherosclerotic cardiovascular risk, moving from a merely glucometabolic approach to a more comprehensive strategy focusing on cardiovascular prevention in the setting of T2D.

In particular, the 2019 European Society of Cardiology (ESC) Guidelines on diabetes, pre-diabetes, and cardiovascular diseases, developed in collaboration with the European Association for the Study of Diabetes (EASD), advise GLP1-RAs in patients with T2D and CV disease or at high CV risk to reduce cardiovascular events (class of recommendation I, level of evidence A) [15]. More recently, the 2021 ESC Guidelines on cardiovascular disease prevention confirmed this indication, suggesting GLP1-RAs in individuals with T2D and atherosclerotic cardiovascular disease (ASCVD) to reduce CV and cardiorenal outcomes (class of recommendation I, level of evidence A) [14].

Although these drugs revolutionized diabetes therapy, their use was limited by several contraindications that should be considered before prescription, such as pancreatitis, pregnancy or breastfeeding, history of previous hypersensitivity reactions, personal or family history of medullary thyroid cancer, and multiple endocrine neoplasias (MEN) [16]. Lastly, administering these drugs by injections may represent another drawback since most patients prefer an oral therapeutic regimen. Conversely, these agents rarely cause hypoglycaemia, representing really safe drugs.

Previous reviews extensively documented the cardiovascular benefit of these agents; most of them focused on clinical evidence from several large randomized trials published in the last ten years on this topic. Differently from previous scientific works [17], in our review, we examine the cardiovascular effects of GLP-1RA and DPP-4i on cardiac outcomes and CV risk reduction, concentrating on the molecular mechanisms they target on immune, endothelial, vascular smooth muscle cells, and platelets, highlighting their importance in atherosclerotic plaque development. In this way, we seek to stimulate a more extensive use of these drugs in clinical practice as helpful cardiovascular preventive strategies.

## 2. Methods

This comprehensive review was realized through literature research from PubMed, Embase, and Cochrane databases. Five reviewers independently searched the literature, including the most relevant clinical trials, pre-clinical studies and reviews published in journals with a high-impact factor. The search lasted until May 2023. Only articles written in English were considered. The keywords used for searching were: “incretins”, “type 2 diabetes mellitus”, “cardiovascular disease”, “atherosclerosis”, “glucagon-like peptide-1 receptor agonist (GLP1-RA)”, and “dipeptidyl peptidase-4 (DPP4)”. Articles were excluded if not published in journals with adequate impact factor or if not relevant to the topic. As a comprehensive and not systematic review, we had no predetermined research questions or specified protocols. Disagreements were resolved by discussion with other team members.

The first section of the review describes the physiological mechanisms of incretins on pancreatic β cells and other cell lines. The second one focuses on the effect of GLP1RA and DPP4i in atherosclerotic process modulation with particular attention to molecular mechanisms in platelets, endothelial, immune, and vascular smooth muscle cells. The third part concentrates on the clinical evidence of DPP4i and GLP1RA efficacy in cardiovascular events prevention, myocardial infarction, and percutaneous coronary intervention.

## 3. Physiological Mechanisms of Incretins

Atherosclerosis is a multifactorial process characterized by forming fibrofatty lesions within the arterial wall and is considered the leading cause of death worldwide [18]. Improvement in treatment and prevention is crucial, especially in patients with T2D, a clinical syndrome expected to affect 783.2 million people by 2045 [19]. Therefore, the treatment guidelines for T2D patients recommend a patient-tailored approach based on lifestyle modifications and the choice of optimal therapeutic option. An ideal anti-diabetic drug should have the following characteristics: significant impact on weight and cardiovascular comorbidities, low risk of hypoglycemia and adverse events, and, last but not least, low costs. Even if no optimal medication exists, incretins represent one of the most attractive and promising options [20]. The “incretin effect” indicates the amplification of pancreatic insulin secretion induced by these gastrointestinal tract-released hormones [21]. Incretins were demonstrated to reduce glucagon concentrations, improve insulin sensitivity, and slow down gastric filling in diabetic patients, with decreased free fatty acid concentrations and body weight. Moreover, beyond glycemic control, incretins protect the cardiovascular system [22]. The glucose-dependent insulinotropic polypeptide (GIP), a 42 amino acid hormone, and glucagon-like peptide-1 (GLP-1), a 31 amino acid hormone, are the most critical studied incretins. They bind to distinct G-protein-coupled receptors highly expressed on pancreatic β-cell surfaces. More specifically, while GLP-1 is secreted by L-cells in the ileum, colon, and rectum, GIP is released from K-cells located predominately in the duodenum and proximal gut after feeding. GIP circulates in 10-fold higher concentrations than GLP-1, whereas GLP-1 appears more potent than GIP. The two hormones are secreted in parallel but are not stored in the same cytoplasmic granules [23]. GIP and GLP1 can be degraded by DPP-4, an amino-peptidase transmembrane protein with a large extracellular domain and a flexible segment anchored in the cell membrane expressed in most cell types. The enzyme is responsible for cleaving and inactivating both two peptides at one of the last alanine residues [24]. Inhibition of DPP-4 and the use of injectable GLP-1RA are the two strategies for potentiate incretin receptor signaling [23]. However, studies on GIP monotherapy were unsuccessful; this could be explained by the fact that, in diabetic patients, the endocrine pancreas remains responsive to GLP-1, but it is no longer responsive to GIP, so this must represent the most likely reason for the reduced incretin action of this hormone [25].

### 3.1. Acute and Chronic Effects of GLP-1 on Pancreatic β Cells

GLP-1 exerts acute and chronic functions on pancreatic cells by binding to its receptor (Figure 1). Acutely, it triggers most insulin release from these cells in a glucose-dependent manner. Hence, glucose enters the pancreatic β-cells through glucose transporter-2 (GLUT2). After phosphorylation, glycolysis, and the mitochondrial tricarboxylic acid (TCA) cycle, glucose determines adenosine triphosphate (ATP) production. Increased intracellular ATP concentration leads to K+ ATP-dependent channel closures with consequent accumulation of K+ ions and membrane depolarization. Thus, cell membrane depolarization causes voltage-dependent Ca2+ channel activation, the influx of Ca2+ ions, and exocytosis of a sub-pool of insulin granules, which contains ∼1–5% of available insulin. This initial rapid process is followed by the remaining entire insulin release mediated by GLP-1 binding to its receptor (GLP-1R), as previously mentioned, allowing the release of ∼95–99% of insulin granules [26]. GLP-1R is a G protein-coupled receptor whose activation induces a rapid increase in c-adenosine monophosphate (cAMP), protein kinase A (PKA), and exchange protein activated directly by cAMP (EPAC) upregulation and increased glucose-dependent insulin release [27]. Further to the stimulation of insulin secretion, the chronic effects of GLP-1RA on pancreatic β cells also consist of the deceleration of mass reduction.

Moreover, GLP-1RA-mediated cAMP elevation is responsible for oxidative stress reduction and trough activation of anti-apoptotic genes for pancreatic β cells survival. The latter process is also stimulated by GLP-1RA-induced secretion of insulin-like growth factor-2 (IGF-2) and expression of its receptor (IGF1-R) in β cells. The IGF-2 binding to its receptor determines an autocrine loop that protects β-cells against apoptosis, inducing proliferation [28,29].

### 3.2. Effects of GLP-1 on Other Cell Lines

GLP-1 performs its function by acting not only on pancreatic β cells, but also on other cell lines (Figure 1). Studies assessing GLP-1RA extra-pancreatic effects reveal that exposure of skeletal muscle cells to these agents increases glycogen synthesis, glycogen synthase activity, and glucose metabolism and inhibits glycogen phosphorylase α activity involved in the breaking up of glycogen in glucose subunits [30]. Moreover, skeletal muscles exposed to increased incretin levels become more sensitive to insulin through more developed angiogenesis and vascularization. Indeed, GLP-1 binds to its receptor, abundantly expressed in endothelial cells, causing microvasculature recruitment in skeletal muscles and better muscle perfusion. Muscle perfusion is also favored via PKA-mediated endothelial nitric oxide synthase (eNOS) activation, which induces vasodilation and consequently increases insulin delivery [31]. These processes allow skeletal muscles to increase basal glucose uptake and storage.

GlP-1 exerts a protective effect on the kidney. It is responsible for inhibiting sodium–hydrogen exchanger 3 (NHE3), an antiporter located on the epithelial cells of the proximal tube that imports sodium ions, simultaneously ejecting hydrogen ions in the proximal tubule lumen. NHE3 inhibition increases natriuresis and diuresis [32,33,34].

The mechanisms of this renal function gain may be direct and indirect. In addition to natriuresis, the first ones include the inhibition of proinflammatory cytokines, adhesion molecules, and profibrotic signaling, as well as the reduction in intraglomerular pressure through the inhibition of protein kinase-C (PKC) and the activation of PKA. The indirect mechanisms regard the benefit that GLP-1R agonists exert on other tissues, including improvement in blood pressure, glucose homeostasis, weight loss, and insulin levels that are beneficial for glomerular filtrate [35].

Furthermore, GLP-1 receptors were demonstrated to be located in many areas of the central nervous system; in particular, in those involved in appetite and gastric motility regulation [36]. GLP-1 is not only produced by alfa and L-cells, but also by neurons. It was proved that neuronally produced GLP-1 is transported to the axon terminals and stored in synaptic vesicles until release into the synaptic cleft, or in case of extra-synaptic release, into the brain parenchyma. Recent pre-clinical studies reported that administering GLP-1RA reduces food intake, determining weight loss in animal models [37]. The underlying mechanism is that L-cells-derived GLP-1 passes through the blood–brain barrier by directly affecting the receptors in the hypothalamic areas responsible for appetite control. GLP-1 is also released into the interstitial space near the site of its synthesis (ileum and colon) and then diffuses locally to act on vagal nerve endings embedded into the gut mucosa [38]. Moreover, the administration of GLP-1 in mice improves learning mechanisms, and the deletion of the gene encoding for GLP-1R is associated with neuron degeneration. Because of this, GLP-1RA was proposed as adjuvant therapy in the treatment of neurodegenerative pathologies such as Alzheimer with encouraging results [39].

In visceral adipose tissue, GLP-1 was demonstrated to reduce volume and alter the composition in vivo [40]. GLP-1 exerts this effect through activation of extracellular signal-regulated kinase (ERK), PKC, and AKT pathways with consequent increased adipocytes apoptosis and reduced pre-adipocytes proliferation [27].

After all, GLP-1 acts directly at the gastrointestinal level, inhibiting gastric emptying and postprandial glycaemic peak reduction due to the slower transit of food from the stomach to the small intestine. The mechanisms are still unknown, but GLP-1R is expressed on the parietal cells of the stomach to indicate a direct action of GLP-1 on their secretion. In addition, vagal denervation was shown to abolish the inhibitory effect of GLP-1 on gastric emptying, suggesting that GLP-1 acts through receptors expressed on vagal fibres that regulate gastric motility [41].

## 4. Effect of GLP1 Agonists and DPP4 Inhibitors in Atherosclerotic Process Modulation

Considering their specific mechanisms of action, it is clear that the main players through which incretins, and thus GLP-1R agonists (GLP-1RA) and DPP4 inhibitors (DPP4i), exert their pro-atherogenic actions are the glucagon-like peptide-1 receptor (GLP-1R) and the degrading enzyme DPP4. GLP-1R is a G-protein coupled receptor with an extracellular and transmembrane domain. It is mainly expressed on the pancreatic β cell surface but also on endothelial cells, smooth muscle cells, macrophages, and monocytes [42]. Its binding to an agonist activates multiple downstream signaling pathways, including PKA/signal transduction and the activator of transcription (STAT), PI3K/Akt, MAPK, and NFkB. Hence, in this section, we will discuss in detail the effects of GLP1 agonists and DPP4-i on platelets, smooth muscle cells, endothelium, and the immune system (Table 1 and Table 2).

### 4.1. Macrophages and Lymphocytes

GLP-1 and DPP4 regulate immune cell functions through various mechanisms.

Macrophages can differentiate into two distinct phenotypes: M1 and M2. M1 macrophages are responsible for inflammatory and pro-atherogenic cytokines production, such as tumour necrosis factor-a (TNF-a), interleukin-1 (IL-1) beta, IL-6, and IL-12. Conversely, M2 macrophages release anti-inflammatory cytokines such as IL-10, TGF-b, IL-4, and IL-13, promoting tissue repair and delaying atherogenesis [43]. The upregulation of the STAT3 signaling is a key step for the differentiation of macrophage towards the M2 phenotype [44]. Liraglutide was demonstrated to induce macrophage differentiation towards the M2 anti-inflammatory phenotype in THP-1 and bone marrow-derived macrophages [45]. More specifically, treatment with this GLP-1RA activates the cAMP-PKA-STAT3 pathway in these cells and, at the same time, downregulates STAT1 activity, which is responsible for the M1 phenotype switch. This evidence is confirmed by the increased M2 markers gene expression (arginine-1, macrophage galectin-1, and mannose receptor-1) on the macrophage surface exposed to this drug [46,47,48].

Furthermore, exenatide was demonstrated to reduce macrophage activation after exposition to lipopolysaccharide (LPS), a major component of gram-negative bacteria that induces acute inflammation. In particular, it activates the cAMP/PKA pathway through the binding to its GLP-1R reducing NF-kB, TNF-α, monocyte chemoattractant protein-1 (MCP-1) levels, and increasing C enhancer-binding proteins (C/EPB) beta synthesis by JNK/p38 induction. Notably, TNF- α and MCP-1 are some of the most important cytokines involved in atherosclerosis progression [49]. Conversely, C/EBP beta has a marked anti-inflammatory effect and induces monocyte autophagy [50]. Another consequence of GLP-1R binding is the upregulation of AMP-activated protein kinase (AMPK), which itself, in turn, contributes to C/EPB beta levels elevation. The final effect is a reduction in pro-inflammatory monocytes with the slowdown of the atherosclerotic process [51].

DPP4i also reduce macrophage activation and chemotaxis. Younis et al. demonstrated that vildagliptin therapy had a suppressive impact on IL-1 production compared to metformin therapy alone in patients with coronary atherosclerosis and T2D [52]. Concordantly, alogliptin was demonstrated to upregulate anti-inflammatory CD163+ macrophages expression with a concurrent reduction in inflammatory macrophages in adipose tissue (CD11b+, CD11c+, and Ly6Chi) [53]. Linagliptin revealed an anti-inflammatory/antiatherogenic effect, increasing insulin sensitivity and decreasing macrophage recruitment in the same cell line [54].

Moreover, incretins influence macrophage chemotaxis by exerting their actions on adenosine deaminase (ADA), an enzyme involved in purine metabolism and whose activation is critical for chemokine-induced inflammatory cell movement [55]. Indeed, while DPP-4 binds ADA, stimulating the chemokine C-C motif ligand 2 (CCL-2 or MPC-1) CCL-5-mediated chemotaxis, and favouring RAC-1-induced movement of monocytes, alogliptin is able to suppress these pathways, reducing monocyte chemotaxis and inflammation in Male LDLR-/- mice [56]. Accordingly, Arakawa et al. demonstrated that GLP-1RA significantly reduced monocyte adhesion to aortic endothelial cells through CD11b in C57BL/6 mice [42].

Finally, incretins were demonstrated to attenuate the switch of macrophages into foam cells, a critical step in atherosclerotic plaque formation. During this process, different scavenger receptors (CD36, SR-A, and Lox-1) exposed on macrophage cell membranes internalize ox-LDL transported through endosomes in the cytosol. Inside the endosomes, lysosomal acid lipase (LAL) converts the cholesterol esters of the ox-LDL into free cholesterol (FC) and free fatty acids (FFA). FC is re-esterified by acyl-coenzyme A: cholesterol acyltransferase-1 and -2 (ACAT1 and ACAT2) and transported in the endoplasmic reticulum, which produces lipid droplets. Through cholesterol efflux transporters, such as ATP-binding cassette (ABCA1, ABCG1) and SR-B1, FC can be excreted by macrophages [57]. Interestingly, liraglutide and exendin-4 were demonstrated to inhibit this process in different ways. Tashiro et al. reported a GLP-1RA-mediated downregulation of the ACAT1 enzyme [58]. Moreover, incretins activate PKA with consequent CD36 scavenger receptor downregulation and higher platelet-activating factor acetyl-hydrolase (PAF-AH) expression, a lipoprotein-associated enzyme that degrades circulating ox-LDL [59].

GLP-1R activation exerts an anti-inflammatory action also on lymphocytes. It was established that GLP-1R-/- mice showed lower levels of circulating CD4+ and CD25+ anti-inflammatory T-reg cells [60]. However, it is DPP4 that plays a major role in lymphocyte regulation. DPP-4 was initially discovered in the immune system and identified as cell surface marker CD26. On T-cells, CD26 expression is generally modest, but rises in response to T-cell activation. In contrast to Th2 clones, lymphocytes with a Th1 phenotype express higher levels of CD26, which increase even more following stimuli that encourage the establishment of Th1 response. The association between CD26 expression and Th1-like immune responses was hypothesized to result from an IL-12-dependent upregulation. In fact, IL-12 increases CD26 expression (DPP-4 activation) on phytohemagglutinin (PHA)-stimulated peripheral blood mononuclear cells [61]. Furthermore, it was suggested that there is a possible role for DPP-4 inhibitor therapy in reducing inflammation linked to T2D and obesity. Indeed, higher HbA1c levels were associated with increased CD26 lymphocytes expression, and DPP-4i reduced CD26 expression on the T-cells surface of diabetic patients [62].

The interaction between DPP4i and other drugs is also important in amplifying their anti-inflammatory actions. In visceral adipose tissue, the third-generation angiotensin receptor blocker (ARB) telmisartan enhances insulin sensitivity and induces macrophage differentiation to the anti-inflammatory phenotype [63]. Co-administration with ARBs strengthens DPP-4i anti-inflammatory and incretin-enhancing activities, potentially improving cardiovascular outcomes [64,65,66].

### 4.2. Endothelium

The endothelium is the continuous cellular lining of the vascular wall’s inner layer. It serves as a physical barrier separating the blood from tissues and is essential for maintaining the right haemostatic balance [67]. Via anticoagulant and antiplatelet pathways, it inhibits thrombosis under physiological circumstances. It also produces nitric oxide (NO) and other substances that support healthy vascular tone, blood pressure, and regional blood flow [68]. On the other side, endothelial dysfunction caused by reactive oxygen species (ROS), apoptosis, loss of barrier function, and switch into mesenchymal phenotype are considered early markers of atherosclerosis [69]. Notably, incretins were reported to modulate different molecular pathways of endothelial cells with a potential protective role against pro-atherogenic processes.

#### 4.2.1. Reactive Oxygen Species and Nitric Oxide Production

Reactive oxygen species (ROS) are among the main causes of endothelial dysfunction, and their production derives from different sources, including nicotinamide adenine dinucleotide phosphate (NADPH) oxidase function, mitochondrial electron transport chain, and eNOS uncoupling.

NADPH oxidase is an enzymatic complex located on the cellular membrane that catalyzes the production of superoxide free radicals (O^2−^) [70]. High glucose levels stimulate PKC, which acts as an NADPH activator determining increased ROS production. Significantly, in human aortic endothelial cells, liraglutide was demonstrated to reduce ROS production by inhibiting PKCb2-mediated activation of NADPH oxidase [71].

Furthermore, another source of ROS comes from the mitochondrial electron transport chain. Mitochondria are cell organelles whose autophagy (mitophagy) plays an essential role in ROS and cell homeostasis maintenance [72]. The process is driven by PTEN-induced putative kinase 1 (PINK1) protein, usually held under a threshold level and degraded by cellular proteases. In conditions of cell stress, PINK1 moves on the mitochondrial external membrane recruiting Parkin, a cytosolic ubiquitin ligase that translocates to damaged mitochondria and promotes their degradation with ROS leakage [73]. In HUVEC, liraglutide acts upstream of this pathway, inhibiting PINK1/Parkin recruitment and preventing excessive mitophagy, thus lowering ROS release [74].

Incretins also exert their endothelial protective effects, influencing the NO production by these cells. NO has a key role in regulating vascular tone, such as endothelial cell proliferation and death [75,76]. The main NO producer is the eNOS, a calcium/calmodulin-dependent phosphoprotein that catalyzes circulating O2 and L-arginine conversion in L-citrulline and NO. In vascular smooth muscle cells (VSMCs), NO promotes guanylate cyclase-mediated cGMP production, stimulating vasodilation. Moreover, NO increases the superoxide dismutase (SOD) activity with a reduction in superoxide anion levels [77]. However, under pathological conditions, the cofactor tetrahydrobiopterin (BH4) is oxidized in dihydrobiopterin (BH2), causing e-NOS uncoupling, and thereby a dimeric enzyme, producing high levels of superoxide anion [78]. Exenatide was demonstrated to increase e-NOS mediated NO production and improve the oxidative stress defense system through the GLP-1R/AMPK pathway stimulation in human subcutaneous endothelial cells [79]. Similarly, exendin-4, always through the GLP-1R/cAMP pathway, activates PI3K/Akt that subsequently upregulates GTPCH1 levels (an enzyme involved in BH4 synthesis), increasing coupled eNOS function [80]. Moreover, GLP-1RA can reduce endothelin-1 (ET-1) production via NF-kB phosphorylation inhibition, leading to increased vasodilation in endothelial cells [81]. In aortic endothelial cells also, the DPP4i sitagliptin was demonstrated to lower ET-1 levels, activating the AMPK pathway responsible for the suppression of the NF-κB/IκBα system [82]. Furthermore, DPP4i improves endothelial function by acting on eNOS. In the study by Shah et al., the administration of alogliptin to C57BL/6 mice with pre-constricted aortic segments led to an increase in NO generation via the Src-mediated PI3K/Akt/eNOS signaling pathway [83]. Concordantly, in aortic and glomerular endothelial cells from obese Zucker rats, treatment with saxagliptin demonstrated an increased NO and reduced peroxynitrite (ONOO-) production [84].

Lastly, another process triggered by ROS overproduction is cellular senescence. Therefore, ROS induces endothelial DNA damage, causing the cessation of cell division. In HUVEC, treatment with exendin-4 has a protective effect on ROS-induced senescence instead. Indeed, GLP1R binding determines the activation of the cAMP/PKA pathway that, via the cAMP response element-binding (CREB) phosphorylation, enhances the heme oxygenase-1 (HO-1) gene expression, preventing H2O2-induced endothelial cell senescence [85].

#### 4.2.2. Apoptosis and Inflammation

GLP-1RA also exerts significant anti-inflammatory and anti-apoptotic effects on the endothelium. In this setting, a key role is played by the NLRP3 inflammasome, composed of three parts: NLRP3, caspase-1, and apoptosis-associated speck-like protein (ASC). Therefore, the activation of the NLRP3 inflammasome translates into IL-1 beta and IL-18 cytokines upregulation and activation of caspase-1, which cleaves gasdermin D (GSDMD), whose N-terminal domains (GSDMD-N) oligomerize and form pores in the cell membrane with consequent cell death through pyroptosis. The inflammasome is activated by endoplasmic reticulum stress-associated protein thioredoxin-interacting protein (TXNIP), NOX-4 (NADPH Oxidase 4) and intracellular ROS, while sirtuin-1 (SIRT1), an important member of the class III histone deacetylase family, negatively regulates it [86].

The GLP-1RA dulaglutide was reported to suppress high-glucose-induced NLRP3 inflammasome activation in HUVECs through TXNIP and NOX4 downregulation and SIRT1 activity increase. This also led to reduced inflammatory cytokine (IL-1 beta, IL-18) and ROS levels [86]. Through the same pathway, liraglutide also reduced intimal hyperplasia after stent implantation [87].

Notably, the anti-inflammatory and anti-apoptotic effects of GLP-1RA also involve endothelial progenitor cells (EPC) responsible for endothelium regeneration. In EPC, the protein SDF-1 (alternatively known as chemokine C-X-C motif ligand 12-CXCL12), via the interaction with the CXCR7, activates p38-MAPK, thus increasing IL-6 production, an interleukin with anti-inflammatory function, and favoring EPC proliferation. Another consequence of the p38-MAPK activation is a significant reduction in cleaved caspase-3 expression, reducing EPC apoptosis.

Thus, in EPCs isolated from rats, exendin-4 was demonstrated to ameliorate EPC survival and function precisely through the previously described pathways [88]. Furthermore, different miRNAs are regulated by GLP-1RA and responsible for endothelial cell survival, particularly miR-93-5p, miR-181a-5p, miR-34a-5p, and miR-26a-5 [89]. Another apoptotic mechanism derives from the activation of the C/EBP homologous protein (CHOP) due to unfolded protein response (UPR). UPR results from unfolded protein accumulation in the endoplasmic reticulum under stress conditions. The cell initially attempts to destroy or refold such proteins and activates cell apoptosis if it fails within a certain amount of time. The proteins that kick off this process are the activating transcription factor 6 (ATF6), the inositol-requiring enzyme 1α (IRE1α), and the phospho-protein kinase R (PKR)-like endoplasmic reticulum kinase (pPERK), located on the endoplasmic reticulum surface and whose activation leads to upregulation, respectively, of ATF6F, X-box binding protein (XBP1), and ATF4 with consequent UPR-related apoptosis [90]. In this regard, exenatide was demonstrated to reduce endothelial reticulum stress apoptosis, and the mechanisms above described by activating the p38/MAPK pathway [91]. Furthermore, liraglutide was shown to suppress IRE1α and PERK activation, as well as ATF6 and chaperone glucose-regulated protein 78 (GRP78) expression in human coronary artery endothelial cells; this is lastly able to activate CHOP through the PERK pathway [92].

#### 4.2.3. Barrier Properties

GLP-1 agonists were also reported to preserve the integrity of the endothelial barrier, composed of endothelial junction and cytoskeleton, whose dysfunction is a critical step in the initiation of the atherosclerotic process [93].

In cultured endothelial cells, exendin-4 binding to GLP-1R activates both c-AMP/PKA and c-AMP/EPAC-1 pathways that increase Ras-related C3 botulinum toxin substrate 1 (Rac1) activity, responsible for the stabilization of cortical actin and reduction in stress fibre formation. EPAC-1 also enhances VE-cadherin internalization, which organizes the opening and closing of the endothelial barrier and is central in permeability changes, thereby contributing to endothelial cell barrier preservation [94]. Notably, advanced glycosylation end products (AGE) also affect endothelial barrier integrity. Via the RAGE stimulation (their receptor), they cause activation of the Rho/ROCK/MYPT and MAPK/MLCK pathways, leading to myosin light chain (MLC) de-phosphorylation and barrier dysfunction. Oppositely, the GLP-1R activation, via c-AMP/PKA triggering, reduces RAGE activity, increasing the levels of phosphorylated MLC, improving cytoskeletal re-organization, and finally decreasing endothelial permeability [95].

Finally, DPP-4i influences the recruitment of EPCs, which is fundamental for membrane damage repair [96,97]. For this purpose, the stromal cell-derived factor-1 (SDF-1) is indeed generated from damaged endothelial tissue; this chemokine, when in the bloodstream, binds the CXCR4 receptor located on vascular and hematopoietic cells and induces bone marrow-derived EPCs mobilization and migration. Notably, SDF-1 is turned inactive by the endothelial transmembrane DPP-4. The inhibition of DPP4 consequently prevents SDF-1 from being degraded and enhances the SDF-1/CXCR4 signaling pathway, increasing the migration of hematopoietic progenitor cells to the site of damage [98,99]. For instance, in the work by Huang et al., sitagliptin increased SDF-1 levels, which then resulted in an increase in circulating EPCs in a GLP-1-independent manner [96,100].

#### 4.2.4. Adhesion Molecules

Another key step for atherosclerosis initiation is the adhesion of macrophages to the endothelial surface with consequent migration into the subintimal layer and switching in foam cells.

Liraglutide reduces vascular cell adhesion molecule-1 (VCAM-1), intercellular adhesion molecule-1 (ICAM-1), plasminogen activator inhibitor type 1 (PAI-1), and P-selectin expression in endothelial cells through the inhibition of NF-kB cascade, translating into reduced leukocyte rolling and vessel infiltration [101]. VCAM-1 and E-selectin expression are also reduced by the Krüppel-like factor 2 (KLF2). Elevated levels of circulating ox-LDL can downregulate KLF2 expression. However, dulaglutide was demonstrated to revert this process, preventing phosphorylation of p53 protein and playing a protective role in preventing adhesion molecules expression [102].

#### 4.2.5. Endothelial Mesenchymal Transition

Some of the processes mentioned above, such as oxidative stress, inflammation, and hyperglycaemia, can induce the endothelial mesenchymal transition (EndMT) process, with endothelial cells that gradually lose their specific proteins, enhancing the expression of mesenchymal-specific genes and favoring neointima formation. The most critical EndMT inducer is the transforming growth factor beta 1 (TGF-β1) with the activation of the Snail–Smad pathway [103]. In HUVECs, liraglutide was demonstrated to interrupt this process preventing Smad2 phosphorylation by activating the AMPK pathway [104].

### 4.3. Vascular Smooth Muscle Cells

Vascular smooth muscle cells (VSMCs) contribute to the atherosclerosis process in different modalities. When stimulated by inflammatory cells and cytokines, they proliferate and migrate from tunica media to the sub-intimal layer. Significantly, this process is amplified under hyperglycaemic conditions due to the activation of the ERK 1-2 and PI3K/Akt pathways [105]. In this regard, both exendin-4 and liraglutide were demonstrated to reduce VSMC proliferation and migration influencing these pathways [106].

Another crucial step is the VSMC phenotypic switching from a quiescent contractile to an activated synthetic cell. The transition is characterized by a lower expression of muscle-specific molecules (α-actin, Myh11, SM22α, and Calponin), whose production is regulated by myocardin, a potent transcriptional coactivator that binds to the serum response factor (SRF), causing transcription of CArG box-containing target genes and switching of the cell towards a contractile phenotype [107]. In this regard, liraglutide was reported to suppress NF-kB-mediated myocardin inhibition, leading to reduced synthetic cell switch [108]. Moreover, some pathways can determine the switch back to the contractile phenotype. For instance, AMPK activates SIRT-1, a NAD+-dependent class III histone deacetylase that regulates the expression of FOXO3a, favoring the differentiation in contractile cells. Exendin-4, through the activation of SIRT1, was reported to determine the re-differentiation of VSMCs [109].

VSMC was also reported to increase ROS production. Lectin-type oxidized LDL receptor 1 (LOX-1) is the primary receptor of ox-LDL in VSMCs, whose activation causes higher ROS levels. Liraglutide was demonstrated to downregulate LOX1 in human VSMCs preventing the process [110]. Furthermore, Nox-1 represents another crucial ROS source in VSMCs responsible for superoxide anion production, activated by Ras-related C3 botulinum toxin substrate (RAC1), a small GTPase involved in the regulation of different cell functions, in turn stimulated by high angiotensin-II levels [111]. Exendin-4, through the angiotensin II-induced cAMP/PKA pathway inhibition, suppresses the action of RAC1 with consequently reduced ROS production and VSMC senescence [112].

Furthermore, VSMCs produce extracellular matrix (ECM) and metalloproteinases (MMPs), MMP-2, and MMP-9 in the atherosclerosis progression phase. These gelatinases are responsible for ECM degradation, but also regulate growth factor production and migration/proliferation of VSMCs in atherosclerotic plaque. MMP production is favored by TNFα-induced phosphorylation of AKT. Exendin-4 was demonstrated to inhibit this process by reducing AKT phosphorylation in the threonine 308 residue in coronary artery VSMCs [113].

DPP4 induces VSMC proliferation after activating different pathways. For that reason, it is not surprising that the vast majority of data suggested that DPP4i directly reduces smooth muscle cell growth. DPP4 serves as a proteinase-activated receptor 2 (PAR2) agonist in VSMCs. PAR2 is a 7-transmembrane receptor, activated by different proteases after proteolytic cleavage of his extracellular N-terminus and consequent presentation of newly exposed tethered ligand (TL). Without proteases, a synthetic peptide that matches the TL of PAR2 selectively activates it. The cysteine-rich region of DPP4 contains this PAR2 activation sequence. When PAR2 is activated, VSMC mitogenesis increases through NF-kB, MAPK, and ERK1/2 pathways. DPP4i alter the conformation of circulating soluble DPP4, preventing its binding to PAR2 and reducing VSMC proliferation [114,115]. A previous study on animals showed that linagliptin decreases VSMC proliferation regardless of its glucose control effect, which may be directly related to accessorial caspase-3-mediated VSMC death [116]. Moreover, activating the nuclear factor erythroid 2-related factor 2 (NRF-2) signaling pathway and suppressing the expression of the chemokines MCP-1 and VCAM-1 are the mechanisms by which gemigliptin has antiproliferative and anti-migratory effects in VSMCs [117]. The DPP4i sitagliptin was reported to reduce platelet-derived growth factor (PDGF)-induced VSMC proliferation in cultured human pulmonary artery smooth muscle cells (PASMCs) via dose-dependently upregulating the phosphatase and tensin homolog deleted on chromosome 10 (PTEN) genes [118]. In another study analyzing diabetic mice treated with vildagliptin for four weeks, there was a significant decrease in the stenosis of damaged carotid arteries in animals receiving this drug compared with the control group. This result was attained by inhibiting VSMC proliferation via the ER stress/NF-B pathway activation [119]. Anagliptin also downregulates proliferation by preventing ERK phosphorylation [120]. Finally, Takahashi et al. recently found that linagliptin and empagliflozin combination therapy considerably moderates VSMC proliferation by reducing VSMC DNA synthesis in vitro [116].

### 4.4. Platelets

The role of platelets in thrombosis and hemostasis is significant [121], and increased levels of coagulation factors, hyperactive platelets, and enhanced surface expression of integrins are all signs of a prothrombic condition, particularly in T2D patients [122,123,124]. GLP-1RA and DPP4i were also shown to act at this level, especially in thrombotic diathesis reduction. About GLP-1RA, exenatide reduces thrombin, ADP, and collagen-induced platelet aggregation in human megakaryocyte cell lines. Moreover, treatment with this drug lessens thrombus formation in a mouse cremaster artery laser injury model [125]. Similarly, through the GLP1-R activation on the platelet surface, liraglutide stimulates the cAMP/PKA/eNOS pathway, leading to better and coupled eNOS function, reduced oxidative stress, and increased NO production. The consequence is diminished endotoxaemia-induced microvascular thrombosis in C57BL/6J mice [126].

About DPP-4i, anagliptin was recently reported to prevent FeCl3-induced arterial thrombus in mice under chronic stress by reversing an imbalance between the disintegrin and metalloproteinase with thrombospondin motifs (ADAMTS13) and the von Willebrand factor (vWF) [127]. Recent research showed that linagliptin dramatically decreases platelet mitochondrial respiration by maintaining cAMP-dependent phosphodiesterase. Lowering the phosphodiesterase activity; it also restricts the decrease in cyclic AMP in thrombin-activated platelets and improves NO capacity to prevent platelet aggregation [128]. Under transmission electron microscopy, platelets pre-treated with linagliptin revealed noticeably reverted morphological alterations when activated by thrombin, including the formation of granules and fewer mitochondria [129]. Moreover, it is well known that platelet tyrosine phosphorylation proteins influence intracellular free calcium levels. In the study by Gupta et al., sitagliptin revealed a high concentration-dependent antiplatelet action due to the inhibitory effect on intracellular free calcium and tyrosine phosphorylation. In this study, thrombin-induced platelet aggregation was followed by increased free intracellular calcium and plasma membrane platelet tyrosine phosphorylation Ca2+-ATPase (PMCA). The administration of sitagliptin significantly decreased platelet aggregation, and in vitro testing verified the same outcomes [130].

## 5. Effects of Incretin-Based Drugs on Lipid Metabolism

Recent evidence shows that incretins and incretin-based therapies positively affect lipid metabolism through the modulation of both lipogenic and lipolytic processes. Specifically, GLP-1 and its agonists inhibit lipogenesis mainly through the cAMP-activated protein kinase (AMPK) pathway, leading to the downregulation of genes involved in the lipogenic process. For instance, a preclinical study conducted by Chen et al. demonstrated that GLP-1 RA significantly reduced the expression of fatty acid synthase (FAS), a key enzyme in de novo lipogenesis [131]. A further preclinical study by Ben-Shlomo et al. confirmed that GLP-1 therapy suppressed lipogenic enzymes such as FAS, stearoyl CoA desaturase-1 (SCD-1), and carnitine palmitoyl transferase-1 (CPT-1) in a mouse model [132]. Similarly, Parlevliet et al. demonstrated that GLP-1RA and DPP-4i inhibited hepatic lipogenesis in high-fat diet mice through AMPK regulation [133]. Incretin-based therapies may also be able to induce lipolysis [134]. In this regard, Xu and colleagues showed that exendin-4 promoted lipolysis in adipocytes through an antioxidant action and the activation of phosphorylated hormone-sensitive lipase (HSL), which is directly involved in lipolysis [135]. Thus, the modulation of lipogenic and lipolytic processes by incretin-based drugs may reduce adiposity and body fat mass, potentially positively preventing dyslipidemia-induced complications. Finally, evidence emerged regarding the anti-lipotoxicity effect of incretins [136] due to several mechanisms such as antioxidant and anti-inflammatory action, increased insulin sensitivity, and improved mitochondrial function and pancreatic cell efficiency [137]. This action may help prevent some disorders due to lipotoxicity, predominantly renal, and cardiovascular complications.

## 6. Effects on Atherosclerotic Plaque Composition and Intimal Hyperplasia in the Pre-Clinical Setting

Several preclinical and experimental studies showed the anti-atherogenic role of GLP-1. These investigations demonstrated reduced aortic macrophage recruitment and atherosclerotic lesion formation in wild-type and apolipoprotein E-deficient mice with infusions of exendin-4, via cAMP/PKA-dependent inflammation suppression, or GLP-1 [138]. Concordantly, Sudo et al. confirmed the beneficial effects of the GLP-1RA lixisenatide on fully developed atherosclerotic plaques performing serial in vivo iMAP-IVUS imaging on brachial arteries of Watanabe heritable hyperlipidaemic (WHHL) rabbits [139]. After 12 weeks of treatment, the percentage of the fibrotic component area was greater, and the percentages of necrotic and calcified areas were lower in animals receiving the GLP-1RA, even without any difference in overall plaque area [139]. Histology from the same plaques analyzed by IVUS indicated less extensive macrophage-positive and calcified deposits in the lixisenatide group than in controls. In contrast, the percentages of VSMCs and fibrotic areas were higher [139].

Incretin treatment is also associated with morphological and compositional characteristics of a potential stable plaque phenotype, as demonstrated by Balestrieri et al. analyzing carotid plaques of symptomatic and asymptomatic diabetic patients undergoing endarterectomy [140]. Hence, GLP-1RA-treated plaques presented more extensive collagen content and sirtuin-6 expression, a transcription factor involved in inflammation and endothelial function, and less inflammatory and oxidative stress [140].

The effects of GLP-1RA in the setting of atherosclerosis progression and percutaneous coronary interventions (PCI) still need to be clarified because of the absence of longer-term studies. There is contrasting evidence about a positive correlation between circulating GLP-1 levels and coronary artery disease (CAD) progression in both diabetic and non-diabetic patients with chest pain undergoing angiography, suggesting even a potential adverse effect of GLP-1 on atherosclerosis [141].

Conversely, recent studies reported the protective role of GLP-1 after myocardial ischemia [142]. Exenatide was demonstrated to reduce coronary artery ligation-induced infarct size in a pig model [143]. Moreover, infarct size reduction was also confirmed in the clinical setting of patients with ST-segment elevation myocardial infarction [144]. Interestingly, GLP-1RA demonstrated protective effects also on neointimal hyperplasia after coronary stent implantation in preclinical studies. In this regard, Xia et al. analysed the effects of liraglutide in diabetic pigs undergoing coronary stenting; angiography and optical coherence tomography were then performed at 22 weeks after stent implantation [125]. On OCT analysis, pigs treated with GLP-1 agonists reported higher lumen area, and lower neointimal thickness and neointimal area. Finally, liraglutide decreased plaque inflammation and significantly lowered diabetes-induced NOD-like receptor family pyrin domain containing 3 (NLRP3), IL-1 beta, and IL-18 expression in stented vascular histological analysis, while it increased expression of IL-10 [87].

Regarding DPP-4i, few studies about PCI are available. Lee et al. showed that nanofiber-eluting stents loaded with vildagliptin were able to accelerate VSMC hyperplasia and stent reendothelialization in diabetic rabbits [145].

In conclusion, these data support a potential intriguing role of GLP-1 receptor activation and DPP-4 inhibition in protecting against atherogenesis, in addition to their established glycaemic actions in the preclinical setting. Of course, long-term studies about the effects on atherosclerotic disease progression are required to investigate their benefit in this clinical setting.

## 7. Cardiovascular Effects of GLP-1RA in Clinical Studies

### 7.1. Effects of GLP-1RA on Cardiovascular Risk Factors

Reductions in systolic blood pressure (BP), lipid levels and body weight were observed in patients treated with GLP-1RA. Since arterial hypertension, dyslipidemia, and obesity are well-known risk factors for atherosclerotic disease, GLP-1RA may reduce atherosclerosis development and progression, adequately improving these parameters. Specifically, the effect of GLP-1RA on BP in previous clinical trials was modest but still correlated to a significant reduction in major cardiovascular events (MACE), with meta-analyses showing that a systolic BP reduction of 10 mmHg was associated with ~20% reduction in MACE [146]. More interesting is the effect on cholesterol levels. For instance, a study conducted by Anholm et al. showed that in patients with stable CAD and newly diagnosed T2D receiving standard statin therapy, liraglutide in combination with metformin improved lipid profile and reduced C-reactive protein (CRP) blood levels [147]. Studies also showed that liraglutide alone was more effective than liraglutide combined with metformin or metformin alone on lipid metabolism and cardiovascular prevention [148].

### 7.2. Effects of GLP-1RA on Cardiovascular Events

Since the Food Drug Administration requested cardiovascular safety data for novel diabetes agents in 2008, numerous trials evaluated the efficacy and safety of GLP-1RA in diabetic patients with cardiovascular disease (CVD). Specifically, eight trials investigated the effects of GLP-1 RA on cardiovascular outcomes (cardiovascular mortality, non-fatal MI, and non-fatal stroke) in patients with T2D and high cardiovascular risk (Table 3) [1,2,3,4,5,6,7,8]. While all these trials demonstrated cardiovascular safety (i.e., non-inferiority) of GLP-1RA, just some of them showed real efficacy in reducing cardiovascular outcomes: the LEADER trial for liraglutide [2], HARMONY outcomes trial for albiglutide [5], REWIND trial for dulaglutide (REWIND) [6], and SUSTAIN–6 trial for injectable semaglutide [3]. Among these, liraglutide was associated with a reduction in both cardiovascular death and all-cause mortality. Moreover, kidney outcomes were improved for liraglutide and injectable semaglutide. Differences across trials may be ascribed to differences across the class of GLP-1RA and differences in the cardiovascular risk profiles of the enrolled patients. A recent metanalysis of these cardiovascular outcomes’ trials by Sattar et al., including 60,080 participants, showed an overall 14% reduction in MACE [149]. Interestingly, another meta-analysis conducted by Gugliano et al. revealed that the beneficial effect of GLP-1RA tends to be greater among subjects with established CVD (HR = 0.84, 95% CI = 0.79–0.90) than among those without CVD (HR = 0.94, 95% CI = 0.83–1.06), although the statistical power was not reached. However, the benefits observed are mainly related to secondary prevention, whereas more studies are needed concerning primary prevention. Lastly, the ongoing SOUL trial will compare the effect on MACE of oral semaglutide versus placebo in subjects with T2D and a high cardiovascular risk profile, trying to fill the existing knowledge gap left by the PIONEER-6 trial in which oral semaglutide vs. placebo failed to reach a statistically significant reduction in MACE (hazard ratio, 0.79 [95% CI, 0.57–1.11]; *p* < 0.001 for noninferiority; *p* = 0.17 for superiority) in a similar cohort of patients [7].

### 7.3. Effects of GLP-1 RAs in the Setting of MI and PCI

As mentioned, emerging evidence supports a beneficial effect of GLP-1RA on MI risk, suggesting a potentially intriguing role of this class of agents in the prompt and effective management of this life-threatening cardiovascular complication. Hence, some clinical studies showed that the use of GLP-1RA in patients presenting with acute MI is associated with a reduction in infarct size (Table 4).

Lønborg et al. demonstrated that acute administration of exenatide in patients presenting with STEMI was associated with a 30% reduction in infarct size [144]. Similarly, Woo et al. demonstrated that exenatide decreased infarct size and improved left ventricular function (evaluated with cardiac magnetic resonance) in STEMI patients undergoing PCI [150].

Another study by Chen et al. confirmed the liraglutide effect on infarct size reduction after STEMI, probably through attenuation of reperfusion injury [151]. The same authors observed that treatment with liraglutide was associated with an improvement in left ventricular ejection fraction (LVEF) 3 months post-revascularization after primary PCI and a reduction in the incidence of no reflow [152,153]. Concordantly, these findings were confirmed by a meta-analysis of randomized controlled trials comparing GLP-1RA with placebo in patients with AMI and undergoing PCI [154]. Recently, exciting data regarding the effect of GLP-1RA treatment in diabetic patients surviving a MI were also collected. In this regard, a prospective observational study included 17 868 patients with diabetes who were discharged alive after a first event of MI, of whom 365 (2%) were using GLP-1RA [155]. The study demonstrated that the use of GLP-1RA, compared with standard diabetes care, was associated with a reduction in the primary composite outcome of stroke, heart failure, re-infarction, or cardiovascular death [adjusted hazard ratio (HR) 0.72; 95%, CI: 0.56–0.92], mainly attributed to a lower rate of re-infarction and stroke [155].

**Table 4 pharmaceutics-15-01858-t004:** Effects of GLP1-RAs and DPP-4i in MI and PCI setting. ↑, increased; ↓, decreased; GLP-1 RAs, glucagon-like peptide-1 receptor agonist; STEMI, ST-segment elevation myocardial infarction; TIMI, thrombolysis in myocardial infarction; PCI, percutaneous coronary intervention; CRP, C-reactive protein; MI, myocardial infarction; ACS, acute coronary syndromes; MACE, major adverse cardiovascular event; CAD, coronary artery disease; and DES, drug eluting stent.

	Molecule	Setting	Main Results
**GLP-1 RA**			
Lønborg et al., 2012 [144]	Exenatide	Patients with STEMI and TIMI flow 0/1 undergoing primary PCI	↓ infarct size (Particularly in those patients with a short duration of ischemia ≤132 min)
Woo et al., 2013 [150]	Exenatide	Patients with STEMI and TIMI flow 0 undergoing primary PCI	↓ infarct size ↑ left ventricular function
Chen et al., 2016 [151]	Liraglutide	Patients with STEMI undergoing primary PCI	↑ myocardial salvage index ↓ infarct size ↓ serum CRP
Chen et al., 2015 [152]	Liraglutide	Patients with STEMI undergoing primary PCI	↑ left ventricular function at 3 months post PCI ↓ no reflow
Trevisan et al., 2021 [155]	GLP-1 RAs	Diabetic patients after a first event of MI	↓ MACE (stroke, heart failure, re-infarction, cardiovascular death)
**DPP4-I**			
Leibovitz et al., 2013 [156]	Sitagliptin	Diabetic patients presenting with ACS	↓ in-hospital complications ↓ 30-day MACE (stent thrombosis, urgent revascularization, post event ischemia, 30-day mortality, re-infarction or re-ischemia, re-admission, stroke/TIA)
Kato et al., 2016 [157]	Alogliptin	Diabetic patients with CAD	↑ coronary flow reserve ↑ left ventricular function
**GLP-1 RA, DPP4-I**			
Santos-Pardo et al., 2021 [158]		Diabetic patients undergoing PCI with DES	No effect on risk of stent thrombosis and intra-stent restenosis.

## 8. Cardiovascular Effects of DPP-4i in Clinical Studies

### 8.1. Effects of DPP-4i on Cardiovascular Risk Factors

Several studies suggest a positive effect of DPP-4i on cardiovascular risk factors. Firstly, DPP-4i exert a favorable action on lipid metabolism. In a meta-analysis of 17 studies, the treatment with DPP-4i was associated with a significant reduction in total cholesterol of about 7 mg/100 mL [159]. DPP-4i also showed important effects in the regulation of BP. Some studies revealed that treatment with sitagliptin and vildagliptin decreases systolic BP independently of a reduction in blood glucose [160,161]. Some other studies revealed that both systolic and diastolic BP were reduced after treatment with vildagliptin [162]. Further, the effect of DPP-4i on inflammation and oxidative stress is also recognized. As proof of this, significant decreases in IL-18 and TNF-α levels were demonstrated in subjects treated with sitagliptin [163]. Moreover, a study conducted by Klempfner et al., including patients with documented CAD, suggested a positive effect of vildagliptin on blood levels of C-reactive protein and MMP-9, which are both inflammation and atherothrombotic markers [164].

### 8.2. Effects of DPP-4i on Cardiovascular Events

Given the anti-inflammatory and anti-atherogenic effects of DPP-4i, this action is expected to translate into clinical benefits regarding cardiovascular outcomes. However, there is contradictory evidence on DPP-41’s impact on cardiovascular events. In this regard, some previous studies demonstrated that diabetic patients using DPP-4i had a lower risk of cardiovascular events than those treated with different anti-diabetic medications [165,166]. Specifically, the meta-analysis by Monami et al. showed that DPP-4i treatment is associated with a reduced risk of MACE with an odds ratio (OR) of 0.689 ([0.528–0.899], *p*  =  0.006), independent of the trial duration, thus suggesting a potential cardioprotective role [166]. Subsequently, five randomized trials were conducted to assess the effects of DPP-4i on cardiovascular outcomes in diabetic patients (Table 5) [9,10,11,12,13]. None of these trials demonstrated a significant benefit on cardiovascular events. Moreover, in the SAVOR-TIMI 53 trial, saxagliptin was associated with an increased risk of hospitalization for heart failure (HF) [148]. Concordantly, in the EXAMINE trial, treatment with alogliptin was associated with a numerical but non-statistically significant increase in HF hospitalization [9]. However, most of these trials were designed to investigate the cardiovascular safety of DPP-4i, and thus the non-inferiority of the experimental treatment versus placebo rather than superiority.

### 8.3. Effects of DPP-4i in the Setting of MI and PCI

Few clinical studies examined the effects of DPP-4i in the setting of MI and PCI, with conflicting results (Table 4). Among these, the SITAGRAMI-trial showed no benefit of therapy with sitagliptin and the granulocyte colony-stimulating factor (G-CSF) on cardiac function and MACE within 12 months of follow-up after a successfully revascularized AMI [167]. In contrast, using data collected by the Acute Coronary Syndrome Israeli Survey (ACSIS) 2010, sitagliptin was demonstrated to determine a lower incidence of 30-day MACE, acute renal failure, pulmonary oedema, and infections among diabetic patients presenting with an acute coronary syndrome (ACS) [156]. Moreover, in a study conducted by Kato et al. the inhibition of DPP-4 by alogliptin improved coronary flow reserve and LVEF in diabetic patients with CAD [157]. Regarding the PCI outcomes, an observational study investigated the association between incretin treatment (GLP-1 RAs and DPP-4i) and the risk of stent failure [158]. The investigators showed that among 18,505 diabetic patients undergoing PCI with a drug-eluting stent, treatment with GLP-1RA and DPP-4i did not reduce the risk of stent thrombosis and intra-stent restenosis.

## 9. Conclusions and Future Prospects

In the last ten years, the use of novel antidiabetic agents such as GLP-1RA and DPP4i for treating T2D patients grew. This is also related to the impressive benefit of these agents on the cardiovascular system, as suggested by several cardiovascular outcome trials. However, in this regard, whether GLP-1RA demonstrated a remarkable reduction in MACE in subjects with T2D and high cardiovascular risk profile, a neutral effect was reported for shorter-acting GLP-1RA (such as lixisenatide) and DPP-4i (such as sitagliptin). Despite these conflicting clinical findings, many preclinical and experimental studies suggest a strong rationale for the use of these molecules as cardiovascular drugs, showing their effects on endothelial, inflammatory, and vascular smooth muscle cells and platelets, translating into significant anti-oxidative, anti-inflammatory, and anti-atherogenic properties. Notably, incretins also demonstrated considerable cardiovascular safety, showing no increased risk of hypoglycemia. In addition, the evidence from some clinical studies supporting a positive effect in the setting of myocardial infarction and percutaneous coronary revascularization represents an important challenge for the expansion of GLP-1 RA e DPP4-i therapeutics in non-diabetic patients other than diabetics. Diabetes and cardiology guidelines already included GLP-1RA, with evidence-based benefits to reduce cardiovascular risk in high-risk individuals with type 2 diabetes, independently from the need for additional glucose control. Demonstrating that these molecules reduce myocardial infarction and cardiovascular death with a favorable benefit/risk profile will probably extend their clinical use. Of course, further studies aiming to examine the role of these agents in cardiovascular settings are strongly needed to identify a new effective strategy for primary cardiovascular prevention.

## Figures and Tables

**Figure 1 pharmaceutics-15-01858-f001:**
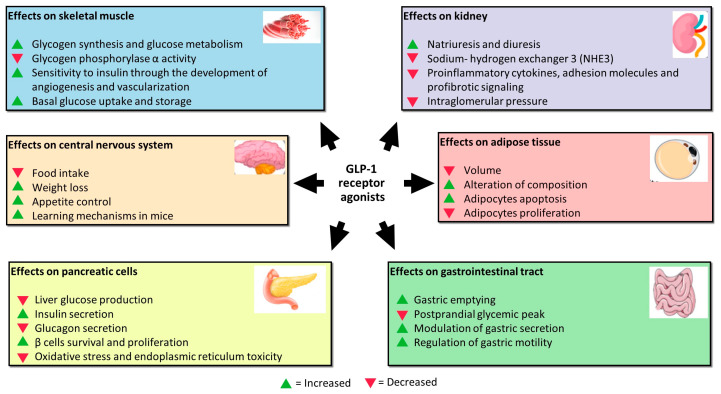
Effects of GLP-1RA on pancreatic β cells and other cell lines.

**Table 1 pharmaceutics-15-01858-t001:** Biochemical effect of GLP1-RA on immune system, vascular smooth muscle cells, endothelium, and platelets. ↑, increased; ↓, decreased; cAMP, c-adenosine monophosphate; PKA, protein kinase A; STAT3, signal transducer and activator of transcription 3; STAT1, signal transducer and activator of transcription 1; M1, classically activated macrophages; M2, alternatively activated macrophages; NLRP3, NOD-like receptor family pyrin domain containing 3; ERK, extracellular signal-regulated kinases; PAR2, proteinase-activated receptor 2; VSMCs, vascular smooth muscle cells; NF-kB, nuclear factor kappa-light-chain-enhancer of activated B cells; TNF-α, tumor necrosis factor-α; MCP-1, monocyte chemoattractant protein-1; ACAT1, acetyl-CoA acetyltransferase 1; PKCb2,protein kinase C beta 2; NADPH oxidase, nicotinamide adenine dinucleotide phosphate oxidase; GLP1R, glucagon-like peptide-1 receptor; AMPK, AMP-activated protein kinase; ET-1, endothelin-1; SDF-1, stromal cell-derived factor 1; CXCR4, C-X-C motif chemokine receptor 4; ROS, reactive oxygen species; NO, nitric oxide; EPC, endothelial progenitor cells; and PAI-1, plasminogen activator inhibitor type 1.

TARGET	MOLECULAR PATHWAY	FINAL EFFECT
**IMMUNE SYSTEM**	↑ cAMP-PKA-STAT3 pathway, involved in M2 polarization	Switch to M2 phenotype
	↓ STAT1 activity, responsible for M1 phenotype switch	
	↑ CD163^+^ macrophages expression and ↓ inflammatory macrophages in adipose tissue	Anti-inflammatory effect
	↓ T-cell CD26 expression	
	NLRP3 inflammasome suppression	
**VASCULAR SMOOTH MUSCLE CELLS**	↓ ERK phosphorylation	↓ VSCM proliferation and migration in atherosclerotic plaque
	↑ PAR2 receptor activation	
**ENDOTHELIUM**	↓ NF-kB, TNF-α and MCP-1	↓ foam cells ↓ ROS production ↑ NO production ↑ vasodilatation ↑ EPC recruitment
	↓ ACAT1 e CD36 scavenger receptor	
	↓ PKCb2-mediated activation of NADPH oxidase	
	↑ GLP1R/AMPK pathway	
	↓ NF-kB and ET-1	
	↑ SDF-1/CXCR4 signaling pathway	
**PLATELETS**	↑ cAMP and PKA activation	Microvascular thrombosis reduction
	↓ plasma fibrinogen and PAI-1	
	↓ CD40 soluble levels	
	↓ inflammatory and thrombogenic gene expression	

**Table 2 pharmaceutics-15-01858-t002:** Biochemical effect of DPP4-I on immune system, vascular smooth muscle cells, endothelium, and platelets. ↑, increased; ↓, decreased; VCAM-1, vascular cell adhesion protein 1; ICAM-1, intercellular adhesion molecule-1; PAI-1, plasminogen activator inhibitor type 1; MMP-2, matrix metalloproteinase-2; MMP-9, matrix metalloproteinase-9; cAMP, c-adenosine monophosphate; PKA, protein kinase A; eNOS, endothelial nitric oxide synthase; cGMP, Cyclic guanosine monophosphate; VASP, vasodilator-stimulated phosphoprotein; PI3-K, phosphoinositide 3-kinases; Akt, serine/threonine kinase; MAPK, mitogen-activated protein kinase; ERK-2, extracellular signal-regulated kinase 2; ADP, adenosine diphosphate; PLT, platelet; ROS, reactive oxygen species; and NO, nitric oxide.

TARGET	MOLECULAR PATHWAY	FINAL EFFECT
**IMMUNE SYSTEM**	↓ VCAM-1, ICAM-1, PAI-1 and P-selectin	↓ leukocyte rolling and vessel infiltration
**VASCULAR SMOOTH MUSCLE CELLS**	↑ MMP-2 and MMP-9	↓ proliferation and migration
	↑ cAMP/PKA pathway	
**ENDOTHELIUM**	c-AMP/PKA pathway activation	Endothelial barrier integrity preservation
**PLATELETS**	↑ cAMP-induced PKA activation	↓ Thrombin-, ADP-, PLT aggregation
	↑ eNOS enzymatic activity	
	↑ cGMP production	↑ NO bioavailability and ↓ ROS production
	↑ VASP-ser239 phosphorylation	
	↓ PI3-K/Akt and MAPK/erk-2 pathway	
	↓ platelet P-selectin expression	

**Table 3 pharmaceutics-15-01858-t003:** Main clinical trials exploring cardiovascular effects of GLP-1RA. GLP-1 RAs, glucagon-like peptide-1 receptor agonist; T2DM, type 2 diabetes mellitus; CVD, cardiovascular disease; MACE, major adverse cardiovascular event; and HHF, hospitalization for heart failure.

	ELIXA [1]	LEADER [2]	SUSTAIN-6 [3]	EXSCEL [4]	HARMONY [5]	REWIND [6]	PIONEER-6 [7]	AMPLITUDE [8]
**Intervention**	Lixisenatide vs. placebo	Liraglutide vs. placebo	Semaglutide vs. placebo	Exenatide vs. placebo	Albiglutide vs. placebo	Dulaglutide vs. placebo	Semaglutide vs. placebo	Efpeglenatide vs. placebo
**Population**	6068 patients with T2D	9340 patients with T2D	3297 patients with T2D	14,752 patients with T2D	9463 patients with T2D	9903 patients with T2D	3183 patients with T2D	4076 patients with T2D
**Established CVD (%)**	100	81	83	73	100	31	85	90
**Follow-up (years)**	2.1	3.8	2.1	3.2	1.6	5.4	1.3	1.8
**MACE**	1.02 (0.89–1.17)	0.87 (0.78–0.97)	0.74 (0.58–0.95)	0.91 (0.83–1.00)	0.78 (0.68–0.90)	0.88 (0.79–0.99)	0.79 (0.57–1.11)	0.73 (0.58–0.92)
**CV death**	0.98 (0.78–1.22)	0.78 (0.66–0.93)	0.98 (0.65–1.48)	0.88 (0.76–1.02)	0.93 (0.73–1.19)	0.91 (0.78–1.06)	0.49 (0.27–0.92)	0.72 (0.50–1.03)
**HHF**	0.96 (0.75–1.23)	0.87 (0.73–1.05)	1.11 (0.77–1.61)	0.94 (0.78–1.13)	0.85 (0.70–1.04)	0.93 (0.77–1.12)	0.86 (0.48–1.44)	0.61 (0.38–0.98)

**Table 5 pharmaceutics-15-01858-t005:** Main clinical trials exploring cardiovascular effects of DPP4 inhibitors. T2DM, type 2 diabetes mellitus; CVD, cardiovascular disease; MACE, major adverse cardiovascular event; and HHF, hospitalization for heart failure.

	EXAMINE [9]	CAROLINA [13]	SAVOR-TIMI 53 [11]	TECOS [12]	CARMELINA [10]
**Intervention**	Alogliptin vs. placebo	Linagliptin vs. Glimepiride	Saxagliptin vs. placebo	Sitagliptin vs. placebo	Linagliptin vs. placebo
**Population**	5380 patients with T2D	6042 patients with T2D	16,492 patients with T2D	14,671 patients with T2D	6979 patients with T2D
**Established CVD (%)**	100	34.5	78.4	100	57
**Follow-up (years)**	1.5	6.3	2.1	3.0	2.2
**MACE**	0.96 (≤1.16)	0.98 (0.84–1.14)	1.00 (0.89–1.12)	0.98 (0.88–1.09)	1.02 (0.89–1.17)
**CV death**	0.79 (0.60–1.04)	1.00 (0.81–1.24)	1.03 (0.87–1.22)	1.03 (0.89–1.19)	0.96 (0.81–1.14)
**HHF**	1.19 (0.90–1.58)	1.21 (0.92–1.59)	1.27 (1.07–1.51)	1.00 (0.83–1.20)	0.90 (0.74–1.08)

## Data Availability

Not applicable.

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
