# Peer review of "Incretins-Based Therapies and Their Cardiovascular Effects: New Game-Changers for the Management of Patients with Diabetes and Cardiovascular Disease"

_pharmaceutics, 2023, doi:10.3390/pharmaceutics15071858_

Round 1
Reviewer 1 Report
The manuscript provides a comprehensive review of literature on “Incretins-based therapies and their cardiovascular effects: new 2 game-changers for the management of patients with diabetes 3 and cardiovascular disease” In fact, the manuscript is well written and provides important information to clearly understand the very nature of incretins and their therapeutic capacity, especially to modulate metabolic processes in pathological conditions. Only a few recommendations to enhance quality. Firstly, it remains important for authors to properly guide the reader on the topics/sub-topics covered within the review. This should be covered within the introduction, especially stating current/available literature (including review articles) on the topic, to indicate how this review is unique. Even though this is an expert/comprehensive review but provide a methodology for study selection. For example, some of the tables discuss prominent clinical trials on the subject, but how was this information searched/selected? This should be clear to the reader. Short falls/recommendations on the use of incretins (especially highlighting the prominent) ones should be covered.
Author Response
We would like to thank the Editor and Reviewers for the careful and thorough reading of our manuscript and for the thoughtful comments and constructive suggestions, which help to improve the quality of this manuscript.
Point 1: firstly, it remains important for authors to properly guide the reader on the topics/sub-topics covered within the review. This should be covered within the introduction, especially stating current/available literature (including review articles) on the topic, to indicate how this review is unique.
Response 1: We thank the reviewer for the comment. In the revised version of the manuscript, we added an introduction paragraph to explain the review's focus and emphasize its uniqueness to the reader. We have also mentioned the most recent literature, including studies, reviews, and guidelines about this topic (page 1, lines 28-63).
Point 2: even though this is an expert/comprehensive review but provide a methodology for study selection. For example, some of the tables discuss prominent clinical trials on the subject, but how was this information searched/selected? This should be clear to the reader.
Response 2: Thanks for the comment. In the new version of the manuscript, we added a paragraph to explain the methodology for study selection and software used for research and bibliography (page 2, lines 66-83).
Point 3: short falls/recommendations on the use of incretins (especially highlighting the prominent) ones should be covered.
Response 3: Thanks for the comment. In this revised manuscript's introduction, we have added indications and contraindications for GLP-1RAs and DPP4-I use proposed by the most recent guidelines (pages 1-2, lines 40-54).

Reviewer 2 Report
Federico Bernardine et al. submitted a very comprehensive review article on the effects of incretins on diabetes mellitus and cardiovascular diseases (CVD). The paper is well written and organized .
There is little objection to the manuscript as a whole except for one issue that might be considered by the authors.
The focus of the manuscript is to a large extent on CVD and there is no doubt that lipids and lipoproteins play a major role in this disease. I therefore believe that the interrelationship of incretins with the lipoprotein metabolism deserves a separate paragraph in this manuscript.
Another possibility might be to insert a table or a cartoon to highlight the features of this topic.
Obviously, if this topic is treated extensively in a separate manuscript of this special issue, this suggestion might be obsolete.
Author Response
We would like to thank the Editor and Reviewers for the careful and thorough reading of our manuscript and for the thoughtful comments and constructive suggestions, which help to improve the quality of this manuscript.
Point 1: The focus of the manuscript is to a large extent on CVD and there is no doubt that lipids and lipoproteins play a major role in this disease. I therefore believe that the interrelationship of incretins with the lipoprotein metabolism deserves a separate paragraph in this manuscript. Another possibility might be to insert a table or a cartoon to highlight the features of this topic. Obviously, if this topic is treated extensively in a separate manuscript of this special issue, this suggestion might be obsolete.
Response 1: We thank the reviewer for the comment. In the new version of the manuscript, we have added a paragraph (N° 3.0) entitled “Effects of incretin-based drugs on lipid metabolism” to explain the effect of incretins and incretin-based therapies on lipid metabolism (lipogenic and lipolytic processes). The most relevant studies in this regard have been cited (page 13, lines 552-573).
